# Human-Alignment and Calibration of Inference-Time Uncertainty in Large Language Models

## Abstract

There has been much recent interest in evaluating large language models for uncertainty calibration to facilitate model control and modulate user trust. Inference time uncertainty, which may provide a real-time signal to the model or external control modules, is particularly important for applying these concepts to improve LLM-user experience in practice. While many of the existing papers consider model calibration, comparatively little work has sought to evaluate how closely model uncertainty aligns to human uncertainty. In this work, we evaluate a collection of inference-time uncertainty measures, using both established metrics and novel variations, to determine how closely they align with both human group-level uncertainty and traditional notions of model calibration. We find that numerous measures show evidence of strong alignment to human uncertainty, even despite the lack of alignment to human answer preference. For those successful metrics, we find moderate to strong evidence of model calibration in terms of both correctness correlation and distributional analysis.

## 1 Introduction

A sizeable body of work has developed around the identification and quantification of uncertainty in the outputs of transformer-based large language models (LLMs). Accurate uncertainty quantification (UQ) is an essential element in predicting model hallucinations and maintaining user trust. In service of that, UQ research has largely focused on developing and utilizing uncertainty measurement methods that are well-calibrated to model accuracy. A well calibrated measure is one that predicts well the model's likelihood of generating a valid answer to the given context. Contexts with high certainty should have a low likelihood of being incorrect and vice versa. A subset of UQ work focuses on measures that are able to be calculated at any time during generation, without additional auxiliary generations. This has often been referred to as inference-time uncertainty quantification. Inference-time measures are uniquely useful in that they can provide a constant signal to the user or to external control modules without significant added computation.

Existing research has not considered whether the investigated UQ measures align with human uncertainty. So, while research has investigated measures with significant calibration, the reported values may not correspond with human uncertainty, making the meaning of the values difficult to parse for users.

Simultaneously, a growing body of research has emerged that seeks to identify human-like behaviors in a variety of LLM tasks and contexts. This has included behaviors as varied as theory of mind (Ullman, 2023; Amirizaniani et al., 2024; Strachan et al., 2024), strategic preferences (Roberts et al., 2024a; Duan et al., 2024), and framing effects (Jumelet et al., 2024; Nguyen, 2024). This work seeks to synthesize these two research thrusts by identifying uncertainty measures that are simultaneously calibrated and aligned to human uncertainty behavior. In particular, we investigate whether any of the uncertainty measures vary consistently with human uncertainty on a per-question basis. Given the difficulty in reliably quantifying uncertainty for an individual human, we approximate this by comparing model measures against disagreement among groups of human survey respondents.

By evaluating LLM uncertainty alignment as well as calibration in inference-time UQ measures, this paper identifies a set of UQ measures which may be effective and more intuitively interpretable by

human users—enabling important advances in LLM-human interaction and prompting the further study of alignment in model signals beyond overt action. This paper specifically contributes to the existing understanding of LLM uncertainty by:

NOTING that top-p selection in LLM decoding is functionally equivalent to the Bayesian highest density credible set, drawing an important but previously un-noted connection between the fields and inspiring our investigation of top-p as a measure of uncertainty.

OBSERVING that many entropy-based inference-time uncertainty measures have *significant* human uncertainty alignment despite *moderate* choice selection and *no* preference ordering alignment.

DEVELOPING a novel ground-truth distributional calibration measure based on shift in the Jensen-Shannon distance metric to directly evaluate the impact of certainty on answer distribution.

SHOWING that the aligned inference-time measures show evidence of calibration in terms of correctness correlation, expected calibration error, and ground-truth distributional calibration.

## 2 PRIOR WORK

Uncertainty Quantification in LLMs is a broad field with numerous notions of uncertainty depending on context, task, and available resources. These are covered in a variety of surveys including Liu et al. (2025); Shorinwa et al. (2025); He et al. (2025). Many of the most successful methods in terms of calibration, like monte-carlo dropout (Shelmanov et al., 2021; Roberts et al., 2024c), rely on multiple generation steps to quantify uncertainty and cannot be readily adapted to quantify per-token inference-time uncertainty levels. Existing work on inference-time uncertainty quantification typically relies on perplexity (Mora-Cross & Calderon-Ramirez, 2024; Margatina et al., 2023; Jiang et al., 2021), maximum token probability (Tian et al., 2023; Steyvers et al., 2025; Huang et al., 2025b; Shrivastava et al., 2023), or entropy methods (Kadavath et al., 2022; Huang et al., 2025b).

Very few works have explicitly investigated the presence of human-like uncertainty responses in LLMs. This work was inspired by preliminary work by Moore et al. (2025), which investigated human-similarity on a diverse set of uncertainty measures. Their work was limited in that the dataset consisted of less than 40 items and did not consider whether the measures which were aligned were also calibrated. We expand on those results by drastically expanding the dataset size and narrow our focus exclusively to inference-time measures. Our work is also novel in that it is the only extant work, to our knowledge, that simultaneously evaluates any uncertainty measures for both alignment and calibration. Other related work, including Argyle et al. (2023); Huang et al. (2025a), have used LLMs and uncertainty-aware procedures to simulate human group responses, but do not seek to establish human-like uncertainty measures.

## 3 INFERENCE-TIME UQ

This work is interested in investigating inference-time UQ methods, as inference-time calculation is necessary for model control mechanisms to have a signal to interpret and react. Non-inference-time methods are valuable tools for diagnostics, model comparison, etc., but are ill-suited for time-sensitive application. While the field of LLM UQ research is large and growing, it has focused on measures of uncertainty that have limited or no inference-time capabilities. These commonly include intuitive methods like self-reporting (Zhou et al., 2023; Mielke et al., 2022; Band et al., 2024; Lin et al., 2022; Tang et al., 2024; Chaudhry et al., 2024; Shrivastava et al., 2023; Tian et al., 2023; Xiong et al., 2023; Belém et al., 2024), multi-inference consistency (Lin et al., 2022; Kadavath et al., 2022; Chen & Mueller, 2024; Manakul et al., 2023; Zhang et al., 2024), and ensemble variation (Wang et al., 2023; Roberts et al., 2024c; Gal & Ghahramani, 2016; Fomicheva et al., 2020). The remainder of this section will describe the inference-time UQ measures employed here. In all cases, these methods are calculated using the token probability distribution over the vocabulary $V$ given some context $c$, $P(v \in V|c)$.

The simplest inference-time UQ measures rely on relative probabilities of the most probable output token (Jiang et al., 2021; Shrivastava et al., 2023; Tian et al., 2023). In this work, we refer to

this simple approach as the top-1 probability. Prior work typically does not promote this as a UQ measure, instead utilizing it as a classifier feature (Jiang et al., 2021) or as a basis of comparison (Shrivastava et al., 2023; Tian et al., 2023).

The majority of our candidate measures are entropy-based measures. These measures are based on the **Shannon entropy** over a probability distribution, $S(P(X)) = -\sum_{x \in X} P(x) \log(P(x))$. Higher entropy distributions are taken to be indicative of higher uncertainty because entropy increases as the relative probabilities throughout the full distribution approach uniform. Typically, this is measured across the entire probability distribution, which we herein refer to as the total entropy for disambiguity.

We further experiment on entropy calculated over a variety of normalized subsets of the total probability distribution. The simplest method of obtaining this subset is using **top-k sampling**, in which the $k$ highest probability tokens are extracted from the total probability distribution. We normalize this subset, $V^k$ by dividing every token probability by the sum of the entire subset. We choose this over softmax because softmax does not necessarily maintain the relative ratio between individual probabilities, which can drastically affect the entropy calculations. We investigate this measure for five values of $k$: 5, 10, 25, 50, and 100. Note that the total entropy is a special case of top-k entropy where $k = |V|$. Because our datasets are exclusively multiple choice format, as are many common benchmarks, we also measure the uncertainty as the entropy over the normalized probabilities of the target tokens, corresponding to the first $n$ letters of the capitalized alphabet, where $n$ is the number of provided answer choices. We call this measure the **choice entropy**.

We also investigate the other common sampling method, top-p sampling. **Top-p sampling**, also known as nucleus sampling, extracts the most probable tokens such that the cumulative probability of the extracted tokens is maximized and less than $p$ and the number of tokens extracted is minimized (Holtzman et al., 2019). Despite its apparent similarity to the highest density credible sets commonly found in Bayesian notions of uncertainty, few prior works have investigated it's viability for LLM UQ. As in Moore et al. (2025), we investigate the size of the resulting token set as a measure of uncertainty. We extend that work by investigating more values for $p$: 0.95, 0.9, 0.75 and 0.5. We also include in our study the entropy over the normalized probabilities of the top-p tokens.

## 4 HUMAN UQ ALIGNMENT

Alignment refers to how closely the behavior of an AI system conforms to the desired behavior of the user or developer. While this is most commonly discussed in terms of how well the models interpret and conform to instructions or moral imperatives, it can also refer to how closely the model's behavior matches human behavior in some context. We define herein uncertainty alignment using the latter notion. That is, *an uncertainty measure is aligned for a given model if the uncertainty measure correlates well with uncertainty in humans*. In this work, we focus on the easier task of correlating with uncertainty among groups of human subjects, as defined by the level of agreement on multiple choice surveys, rather than attempting to measure correlation at the individual level, though this should be explored in future work.

### 4.1 DATASET

We use two datasets to investigate UQ alignment. The first is the dataset compiled in the inspiring work by Moore et al. (2025). This dataset is comprised of 38 manually collected and formatted questions originally sourced from Pew Research Center (2025) surveys . This dataset is clearly limited in size and thus diversity, but provides a useful baseline for comparison.

The second dataset is a collection of 2998 randomly selected questions obtained from the Roper Center for Public Opinion Research (2025) database. The exact methods used to sample from this database are detailed in the appendix. All questions were obtained from human surveys performed during the years 2017-2023. Minor keyword filtering was employed to reduce the number of time-sensitive and personal experience questions. A total of 30571 questions were initially retrieved from Roper. We removed questions with invalid response ratios and sampled 3000 questions from the resulting set uniformly at random without replacement. Each question's answer choices were shuffled to reduce ordering biases, but every model was presented with the same answer choice

ordering for consistency. After removal of two additional questions for invalid answer choice counts, the final dataset was comprised of 2998 questions with an average of 3.77 choices per question.

## 4.2 METHODS

Following is a question and a selection of answer choices. Provide the label for the answer with which you most agree.
Question: <Q-TXT>
   <ANS-LAB-0>. <ANS-TXT-0>
   <ANS-LAB-1>. <ANS-TXT-1>
   . . .
Answer:

Figure 1: Prompt template for querying answer preference. We query the model for the full vocabulary probability distribution, from which we extract the chosen answer and uncertainty metrics.

Each model was queried for the full token probability distribution using a consistent prompt template, described in Figure 1. In addition, we use the standard cloze test to determine the model's chosen answer from the options provided. We extract from the probability distributions each of the uncertainty measures described above in the inference-time uncertainty section. Human group uncertainty is obtained by taking the entropy over the response percentages after normalization.

Across all questions, we measure human agreement in three ways. First, we measure overt agreement based on the response ratios and cloze test results to show how often the models and humans agreed on the best answer. For a more fine-grained analysis, we measure the relative preferential alignment between model and human using the normalized Kendall $\tau$ distance (Kendall, 1938). This measures the minimum number of pairwise swaps needed to convert the model preference order into the human preference order, normalized by the maximum possible distance. Finally, we measure the correlation across all questions between the human group uncertainty and each of the model uncertainty measures.

## 4.3 RESULTS

| Model | LLaMa 3.2 1B | 1B Ins | LLaMa 3.2 3B | 3B Ins | Mistral 0.1 7B | 7B Ins | Mistral 0.3 7B | 7B Ins | LLaMa 3.1 8B | 8B Ins |
|---|---|---|---|---|---|---|---|---|---|---|
| Top Answer Agreement | 0.271 | 0.313 | 0.319 | 0.372 | 0.346 | 0.360 | 0.350 | 0.392 | 0.362 | **0.427** |
| Norm. Kendall $\tau$ Distance Mean | 0.486 | 0.446 | 0.511 | 0.484 | 0.457 | 0.496 | 0.463 | **0.441** | 0.486 | 0.477 |
| Norm. Kendall $\tau$ Distance Std. | 0.339 | 0.350 | **0.337** | 0.340 | 0.349 | 0.338 | 0.349 | 0.346 | 0.343 | 0.343 |

Table 1: Results of preference alignment analysis. All models beat random chance ($\sim 0.265$) at matching human chosen answer, with clear trends on model size and instruction fine-tuning. All models, with no discernable trend, show effectively no downstream preference alignment as defined by Kendall $\tau$ distance.

Table 1 displays the results of the overt agreement and order preference analysis. For overt agreement, models show mild agreement, with all models other than Llama-3.2 1B significantly ($p < 0.01$) beating the theoretically determined random chance ($\sim 0.265$) based on a one-sided one-proportion z test (Moore et al., 2016). The test is appropriate as only one proportion is used given the precise random proportion is calculated and the proportion is from a binomial distribution.

The relational analysis shows remarkably little agreement between model and human. Every model shows a consistent mean distance of $\mu = 0.476 \pm 0.035$ and standard deviation $\sigma = 0.343 \pm 0.006$, indicating effectively random and widely distributed distance scores. Together, these indicate that the models show moderate agreement with humans on the top token, but not on overall token

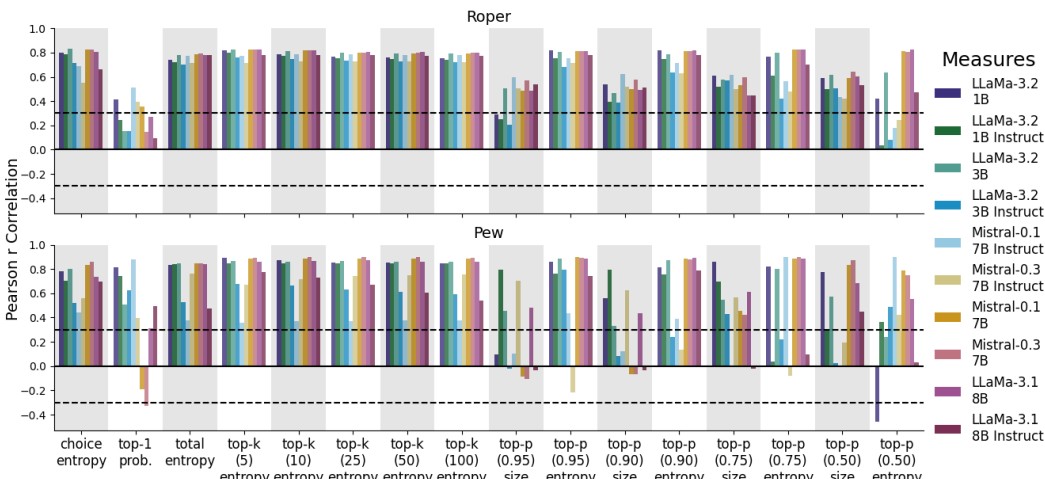

Figure 2: Pearson correlations between human uncertainty level and model uncertainty per model and uncertainty measure. Dotted lines represent a significance threshold of $|r| >= 0.3$. Top: Results for iRoper dataset ($n = 2998$). Bottom: Replicated results for pew dataset ($n = 38$).

preference ordering. This finding is at odds with previous work on strategic preference ordering in LLMs (Roberts et al., 2024b), with the primary difference in our approaches being the prompt design and output capture approach. The prior work used a prompting strategy dubbed counterfactual prompting as an alternative to the more common cloze testing used here. In the prior, the output is measured using a consistent canary token whose probability is queried once for each option while in the latter a set of options is presented once and the relative probabilities associated with each option is taken as the relative preference. This might suggest that model alignment is sensitive to specific prompting methodology and preference interpretation. The lack of preference ordering alignment could additionally be explained by the fact that models are trained only on individual target tokens and have no information about what tokens would have been valid but less preferred tokens.

The results for uncertainty alignment, shown in Figure 2, are much more promising. As in the prior work, we see wild variation in correlation across models for every measure. Even still, every model shows significant ($|r| \geq 0.3$) correlation for all top-k measures, including total entropy. The same is true of choice entropy and top-p entropy for all $p \geq 0.75$, though with progressively more instability as $p$ decreases. Top-p size, counter to previous studies, shows weak but significant correlation for all $p \leq 0.9$. The only measures that do not show consistent significant correlation are top-1 probability, top-p (0.95) size and top-p (0.5) size. It is noteworthy, but currently unexplained that degradation trends are reversed between top-p size and entropy. The top performing measures are identified as those whose correlation exceeds $r \geq 0.5$ for every model. This set includes choice entropy, total entropy, all top-k entropies, and top-p entropy for $p \geq 0.9$. These models are further evaluated for calibration. It should be noted that all of the top performing measures are significant for all models on both datasets, with the exception of those based on top-p entropy.

## 5 CALIBRATION

Calibration is the standard measure by which uncertainty measures are evaluated for LLMs. It refers to the measure's utility in predicting the model's likelihood to correctly complete some task. A well-calibrated uncertainty measure should be low when the model is highly likely to answer correctly and it should be high when the model's likelihood to answer correctly is low or near random chance. We measure calibration on the common mulitple choice question answering benchmark, MMLU (Hendrycks et al., 2020). Note that, unlike the Roper and Pew datasets, each MMLU question always has a constant four available answer choices.

## 5.1 METHODS

Similar to alignment evaluation, each of the MMLU questions are presented to the model using the same prompt template in Figure 1. For each question, the full token probability distribution and cloze test results are recorded. Analysis is split into two phases. In the first phase, a simple measure of calibration is obtained by taking the Spearman correlation between the binary correctness of the cloze test result and the candidate uncertainty measures. Because results can vary wildly within a single model across the various question subjects, we separate by subject during analysis.

### 5.1.1 EXPECTED CALIBRATION ERROR

Expected Calibration Error (ECE) (Guo et al., 2017) is a common measure of calibration that measures the average disparity between model confidence and task accuracy. This is calculated as $\sum_{b \in B} \frac{|b|}{N} |conf_b - acc_b|$, where $B$ is a set of equal width bins into which the $N$ task instances are separated based on certainty level, $conf_b$ is the average model confidence for instances in bin $b$, and $acc_b$ is the observed model accuracy for instances in bin $b$.

All measures investigated in this experiment are entropy based. While normalized entropy can be interpreted as a percentage, this percentage is not directly interpretable as a probability of accuracy. For instance, it is conceivable that a model may have consistently low normalized entropy for all outputs while having a strong calibration signal within the observed range of entropy values as the entropy-based certainty signal may have relative rather than global interpretability. To accommodate this, we assume the observed entropy values range from full uncertainty to full certainty. For each model and measure pair, we standardize the observed uncertainty measure values, $H$, to have a mean of 0 and a standard deviation of 1 to preserve the relative distribution shape. We bin the standardized instances, $H'$, into 10 equal width bins across the range $[-\beta, \beta]$ where $\beta = \max(|\min(H')|, |\max(H')|)$. For a given bin, we take $conf_b = 1 - CDF_{H'}(\mu_b)$ where $CDF_{H'}$ is the observed cumulative distribution function over $H'$. We subtract from 1 as entropy in an ideal, calibrated model should be inversely related to certainty. We use this value to calculate the ECE as described above.

### 5.1.2 JENSEN-SHANNON DISTANCE SHIFT

We provide a more nuanced analysis of the calibration using the shift in Jensen-Shannon distance. The Jensen-Shannon distance (JSD) is a symmetric and finitely-valued extension of the Kullback-Leibler divergence. It is defined as $JSD(P||Q) = \sqrt{\frac{1}{2}D(P||M) + \frac{1}{2}D(Q||M)}$, where $D$ is the K-L divergence function and $M$ is a mixture distribution of $P$ and $Q$. As JSD is a metric, and thus obeys the triangle inequality, we can use it to directly compare the relative distance between two separate pairs of probability distributions over a shared outcome space (Osán et al., 2018). That is, if $JSD(P, P') < JSD(Q, Q')$ and if $P$, $P'$, $Q$, and $Q'$ represent probability distributions over an identical outcome space, this indicates that $P$ and $P'$ are more similar to each other than $Q$ is to $Q'$. This metric thus provides a similar, but more nuanced, view of the model accuracy, but allows for robust hypothesis testing. Our hypothesis is that a well-calibrated measure should show a significant change in JSD from the correct distribution when the model goes from certain to uncertain.

We leverage the JSD to examine how closely the distribution of answers given by the model matches the distribution of correct answers at high and low entropy. For each uncertainty measure, we standardize the measured certainties to have a mean of 0 and standard deviation of 1. We assign the questions to high or low certainty by whether the standardized uncertainty is above or below 0. Questions with a standardized uncertainty of 0 were randomly assigned. From this assignment, we obtain four distributions, $H_A$, $H_M$, $L_A$, and $L_M$. $H_A$ is the count of each correct answer choice for the questions with high certainty, while $H_M$ is the distribution of answers given by the model for those same questions. $L_A$ and $L_M$ are defined similarly. Finally, we use a permutation test with random uncertainty level assignments to test whether $JSD(H_M, H_A) > JSD(L_M, L_A)$ to a significant degree. We dub the resulting calibration metric, $JSD(H_M, H_A) - JSD(L_M, L_A)$, as the JSD shift. We run the permutation test with 1000 permutation iterations.

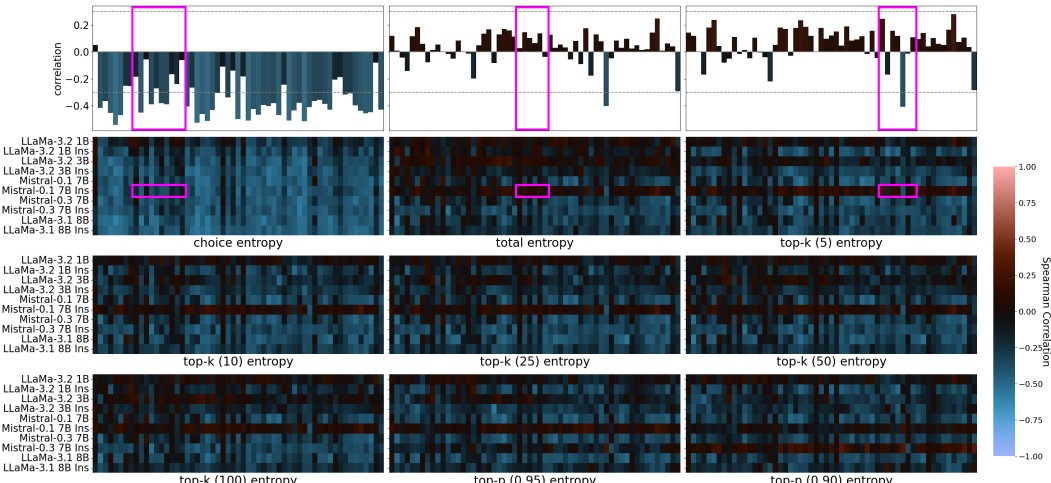

Figure 3: Bottom: Heatmaps showing correlation between uncertainty measurements and correlation per MMLU subject. Each heatmap corresponds to a single measure. In each heatmap, every row represents a model and every column represents one subject. Negative correlation (blue) indicates the model was more likely to be correct as uncertainty decreased. Top: A single model slice of the top row of heatmaps, intended for ease of illustration.

## 5.2 RESULTS

The results of the simple correlation analysis is shown as a heatmap in Figure 3. The ideal measure would appear as bright blue (negative correlation) for all models and subjects. Unlike the alignment case, there is a clear winner in the choice entropy. Across nearly all models and subjects, choice entropy shows mild to moderate correlation with correctness. The primary exceptions are LLaMa 1B, which shows low correlation on all subjects, and a small collection of subjects that show no correlation for any model. In both cases, these are likely indicative of poor performance due to underpowered model or excessive question difficulty.

Outside of choice entropy, all models, with the notable exception of Mistral 0.1 7B Instruct, show negative correlation in most subjects, in particular for top-$k$ entropy. Qualitatively, it appears that there is a non-linear relationship between calibration and size of $k$. The calibration appears to peak

| | choice entropy | total entropy | top-k (5) entropy | top-k (10) entropy | top-k (25) entropy | top-k (50) entropy | top-k (100) entropy | top-p (0.95) entropy | top-p (0.90) entropy |
|---|---|---|---|---|---|---|---|---|---|
| LLaMa-3.2 1B | 0.25 | 0.26 | 0.27 | 0.24 | 0.24 | 0.24 | 0.24 | 0.27 | 0.22 |
| LLaMa-3.2 1B (Ins) | 0.14 | 0.25 | 0.13 | 0.15 | 0.19 | 0.21 | 0.22 | 0.18 | 0.15 |
| LLaMa-3.2 3B | 0.073 | 0.22 | 0.19 | 0.16 | 0.18 | 0.2 | 0.21 | 0.23 | 0.2 |
| LLaMa-3.2 3B (Ins) | 0.097 | 0.23 | 0.13 | 0.14 | 0.17 | 0.19 | 0.2 | 0.14 | 0.14 |
| Mistral-0.1 7B | 0.07 | 0.21 | 0.12 | 0.15 | 0.17 | 0.19 | 0.19 | 0.11 | 0.12 |
| Mistral-0.1 7B (Ins) | 0.084 | 0.28 | 0.3 | 0.29 | 0.29 | 0.29 | 0.29 | 0.27 | 0.14 |
| Mistral-0.3 7B | 0.067 | 0.2 | 0.14 | 0.14 | 0.17 | 0.18 | 0.19 | 0.12 | 0.13 |
| Mistral-0.3 7B (Ins) | 0.12 | 0.13 | 0.12 | 0.11 | 0.11 | 0.12 | 0.12 | 0.13 | 0.25 |
| LLaMa-3.1 8B | 0.08 | 0.14 | 0.17 | 0.12 | 0.12 | 0.12 | 0.12 | 0.14 | 0.19 |
| LLaMa-3.1 8B (Ins) | 0.15 | 0.17 | 0.15 | 0.15 | 0.16 | 0.16 | 0.17 | 0.16 | 0.17 |
| **Average** | 0.11 | 0.21 | 0.17 | 0.17 | 0.18 | 0.19 | 0.2 | 0.18 | 0.17 |

Figure 4: Results of expected calibration error analysis with 10 bins, separated by model and UQ measure. Brighter (lower) values indicate stronger calibration. Choice entropy shows the strongest and most consistent calibration, consistent with the correlation analysis in Figure 3. Also consistent with the correlation analysis, all models except LLaMa 3.2 1B and Mistral 0.1 7B Instruct show moderate to high calibration for nearly all measures.

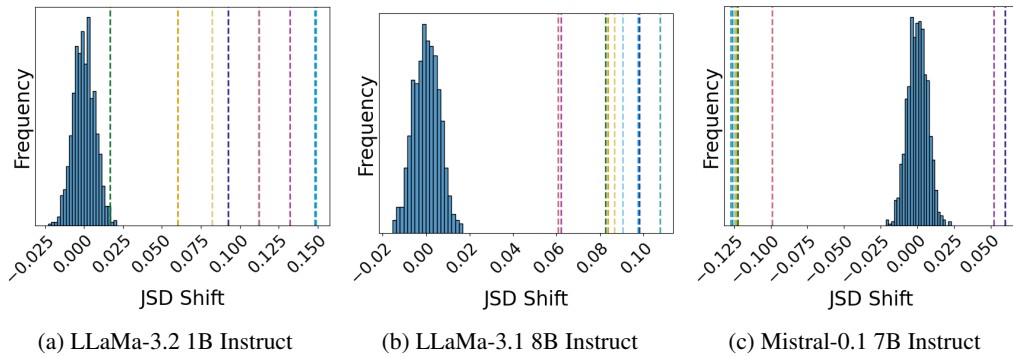

(a) LLaMa-3.2 1B Instruct      (b) LLaMa-3.1 8B Instruct      (c) Mistral-0.1 7B Instruct

Figure 5: Results of permutation testing of JSD shift on LLaMa 3.2-1B Instruct, LLaMa-3.1 8B Instruct, and Mistral-0.1 7B Instruct. Histogram bars represent distribution of JSD shift values for random partitions. Each dotted line represents the observed JSD shift for one measure. Results for all models can be found in Appendix B.

at $k = 10$ and degrades as $k$ increases. Mistral 0.1 7B Instruct is noteworthy in showing weak positive correlation for all measures except choice entropy.

Our analysis shows that all models and measures show mild to moderate calibration in terms of ECE, as shown in Figure 4. In agreement with the previous analysis, we see the highest calibration from choice entropy, with lower but moderate calibration for all other metrics. Of the choice-independent metrics, top-k (5), top-k (10), and top-p (0.9) show the lowest ECE. As before, Mistral-0.1 7B Instruct and LLaMa-3.2 1B are noticeable outliers, with moderately high error across most metrics.

The results of the permutation test on the JSD shift test for a selection of models is shown Figure 5, with the JSD shift and significance for every model-measure pair listed in Table 2. Each dotted vertical line is one uncertainty measure's JSD shift. Figure 5a displays results for LLaMa-3.2 1B Instruct, which shows the weakest observed JSD shift for any metric (intersecting dotted line). Figure 5b, for LLaMa-3.1 8B Instruct, is among the strongest results, with results for most models falling between these two models. A noteworthy exception is Mistral-0.1 7B Instruct (Figure 5c), which shows significant but negative shift for most metrics. This implies, that those metrics may be anti-calibrated for this model, possibly explaining the mixed results for this model in Figures 3 and

| Measure | | LLaMa 1B | LLaMa 1B (I) | LLaMa 3B | LLaMa 3B (I) | Mistral 7B 0.1 | Mistral 7B 0.1 (I) | Mistral 7B 0.3 | Mistral 7B 0.3 (I) | LLaMa 8B | LLaMa 8B (I) |
|---|---|---|---|---|---|---|---|---|---|---|---|
| choice | JSDS | 0.055 | 0.093 | 0.150 | 0.031 | 0.098 | 0.060 | 0.075 | 0.091 | 0.217 | 0.098 |
| | p | 0.000 | 0.000 | 0.000 | 0.000 | 0.000 | 0.000 | 0.000 | 0.000 | 0.000 | 0.000 |
| total | JSDS | 0.034 | 0.017 | 0.040 | 0.020 | 0.018 | -0.122 | 0.023 | 0.085 | 0.144 | 0.082 |
| | p | 0.000 | 0.009 | 0.000 | 0.000 | 0.002 | 0.000 | 0.000 | 0.000 | 0.000 | 0.000 |
| top-k 5 | JSDS | 0.055 | 0.149 | 0.094 | 0.043 | 0.051 | -0.127 | 0.026 | 0.088 | 0.052 | 0.108 |
| | p | 0.000 | 0.000 | 0.000 | 0.000 | 0.000 | 0.000 | 0.000 | 0.000 | 0.000 | 0.000 |
| top-k 10 | JSDS | 0.069 | 0.148 | 0.126 | 0.040 | 0.038 | -0.126 | 0.023 | 0.085 | 0.142 | 0.097 |
| | p | 0.000 | 0.000 | 0.000 | 0.000 | 0.000 | 0.000 | 0.000 | 0.000 | 0.000 | 0.000 |
| top-k 25 | JSDS | 0.061 | 0.112 | 0.099 | 0.035 | 0.033 | -0.125 | 0.022 | 0.083 | 0.158 | 0.091 |
| | p | 0.000 | 0.000 | 0.000 | 0.000 | 0.000 | 0.000 | 0.000 | 0.000 | 0.000 | 0.000 |
| top-k 50 | JSDS | 0.053 | 0.082 | 0.071 | 0.031 | 0.029 | -0.124 | 0.023 | 0.084 | 0.154 | 0.086 |
| | p | 0.000 | 0.000 | 0.000 | 0.000 | 0.000 | 0.000 | 0.000 | 0.000 | 0.000 | 0.000 |
| top-k 100 | JSDS | 0.046 | 0.060 | 0.055 | 0.028 | 0.026 | -0.123 | 0.023 | 0.083 | 0.148 | 0.084 |
| | p | 0.000 | 0.000 | 0.000 | 0.000 | 0.000 | 0.000 | 0.000 | 0.000 | 0.000 | 0.000 |
| top-p 0.95 | JSDS | 0.016 | 0.112 | 0.076 | 0.046 | 0.082 | -0.099 | 0.041 | 0.063 | 0.160 | 0.061 |
| | p | 0.002 | 0.000 | 0.000 | 0.000 | 0.000 | 0.000 | 0.000 | 0.000 | 0.000 | 0.000 |
| top-p 0.90 | JSDS | 0.132 | 0.000 | 0.118 | 0.038 | 0.087 | 0.052 | 0.032 | 0.022 | 0.089 | 0.062 |
| | p | 0.000 | 0.000 | 0.000 | 0.000 | 0.000 | 0.000 | 0.000 | 0.001 | 0.000 | 0.000 |

Table 2: Observed JSD shift values for each model-measure pair, with associated p-values. All values rounded to nearest thousandths place. Only 4 pairs, highlighted in yellow, show $p \geq 0.001$. The top three highest average absolute JSD shift is observed for choice entropy ($\mu_{|x|} \approx 0.0968$), top-10 entropy ($\mu_{|x|} \approx 0.0894$), and top-25 entropy ($\mu_{|x|} \approx 0.0820$). The smallest average absolute JSD shift is observed in total entropy ($\mu_{|x|} \approx 0.0584$). (I) indicates instruction fine-tuned variants.

4. This further suggests that the model itself may be unusually negatively calibrated as compared to the other models tested. Permutation graphs for all models can be found in Appendix B.

# 6 CONCLUSION

Understanding and disclosing the uncertainty of language models is paramount to their trustful use in many applications. However, if the disclosed uncertainty does not correspond to human uncertainty, these measures may undermine the interaction. Further, if the method for measuring the uncertainty requires significant computation, this only increases the considerable impact of model use.

In this work, we found strong initial evidence that many low-computation, inference time uncertainty measures are well-aligned to human uncertainty. This is in spite evidence that the human groups and models rarely agree, both in chosen answer and in answer preference ordering. We identify 9 measures that show especially strong correlation with human group uncertainty: choice entropy, top-k entropy (for numerous values of k), and top-p entropy (for high values of p). Further, we find that these candidate measures show statistically significant calibration on the MMLU benchmark, suggesting these as viable measures of human-aligned, inference time LLM uncertainty.

## 6.1 FUTURE WORK

Future work should seek to find measures that are more highly calibrated without sacrificing alignment. Additional fruitful lines of research could include extending the current research into open-ended contexts—including measures similar and dissimilar to the proposed framework below—and investigating whether uncertainty-aware applications based in human-aligned measures show benefits to user experience or task efficacy by directly conducting human studies. Our work also did not investigate a direct relationship between uncertainty and human-LLM answer agreement, in particular whether answer agreement correlates with uncertainty level. Future work should investigate this relationship to a finer degree. Finally, future work should seek to measure uncertainty alignment at the individual level as well as at the human group level.

The most important area of future work is the introduction of highly calibrated and aligned measures—with top-k 10 being the most promising—into agentic and cooperative software to provide users with an intuitive understanding of model confidence. We hope that such a method will increase trust and improve outcomes in human-LLM cooperation contexts.

## 6.2 A PROPOSED CONCEPTUAL FRAMEWORK FOR OPEN ENDED QUESTIONS

This paper investigated and found human aligned uncertainty measures in a multiple choice context. However, the measures and approach used here can be extended to open answer generation contexts. Consider a language model asked an open ended question. The model generates a response, $R$. The model context including $R$ is then extended with a prompt to the effect, "Evaluate if the answer given is a true, false, or neutral response to the the question". This prompt converts the open ended question response uncertainty problem into a multiple-choice, 3 answer choice question—similar to the 3.77 answers per question on average in the dataset used to establish alignment. The measures evaluated in this paper are then directly applicable with little inferential cost as the precomputed token values do not need to be re-calculated as is necessary in other UQ measure approaches discussed in the related work section. This framework is planned to be evaluated in future work.

# 7 LIMITATIONS

The most significant limitation to this study is that the experiments were limited to only multiple-choice question contexts. Further research is needed to determine whether the results herein will persist in more open ended contexts. Our work is also limited to a selection of widely used open-weight models with 8 billion or fewer parameters. We are prevented by available compute resources from extending our experiments to larger models, but our results do not show any apparent size dependence for the most aligned and calibrated measures. While we did not find a model size dependence in top-k measures, this should be evaluated in significantly larger models to determine the relevance to more powerful large language models.

## 8 REPRODUCIBILITY STATEMENT

All experiments were performed using the high-performance computing resources provided by IN-STITUTION REDACTED. Specifically, we leveraged one A100 GPU with 40GB of VRAM. The code was developed in a Python 3 Jupyter environment using the Hugging Face toolbox (Lhoest et al., 2021). All random processes used predefined seeds that are included as default parameters to the associated functions. Experiments were performed on both the completion and instruct fine-tuned versions of the following models: LLaMa-3.2 1B and 3B (Meta AI, 2024), LLaMa 3.1 8B (Grattafiori et al., 2024), and Mistral 7B versions 0.1 and 0.3 (Jiang et al., 2023). All experimentation and analysis source code has been released under the MIT license and is publicly available at LINK REDACTED. The Pew (Pew Research Center, 2025) and Roper (Roper Center for Public Opinion Research, 2025) datasets are copyrighted and were accessed through the official portals. Therefore, we are unable to independently release the dataset. Our Roper data collection process can be replicated as detailed in Appendix A.

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

## A    HUMAN COMPARISON DATASET CONSTRUCTION

The Roper dataset is comprised of 2998 questions obtained from the Roper Center for Public Opinion Research iPoll database (Roper Center for Public Opinion Research, 2025). These questions were sampled uniformly at random from an initial set of 30571 questions pulled from the database. The exact search criteria used to obtain these questions can be found in Figure 6. The initial set was comprised of all results of this search.

Before sampling, the questions were processed for validity and removal of non-response answer choices. Many questions included non-response answer choices for respondants that had no opinion or preferred to not answer the given question for any reason. We decided to remove these, given that the goal of this research is to investigate the behavior of the model when making a decision, not how likely it is to refuse to make a decision. The latter is a well-established behavior (Arditi et al., 2024), and likely affected by uncertainty, but is outside the scope of this work. We also elected to not remove "None of the Above" and similar answers that indicate an active rejection of the given answers rather than a passive lack of preference. This is consistent with the questions pulled from Roper iPoll, many of which included both rejection and refusal options. Refusal answer choices were found using manual inspection of the dataset, identifying the following refusal options:

- don't know/refused
- don't know/skipped
- don't know/skippedrefused
- no answer
- not selected
- not selected/no answer
- not sure/refused
- not sure/skipped
- omit

> **Exclude:** today OR time OR now OR ahead OR next OR approve OR vote
> **Interview Start:** 01/01/2017
> **Interview End:** 12/31/2023
> **Country:** United States
> **Contents:** Downloadable Datasets
> **Topic:** Economic Issues/Policies OR Entertainment, Arts, and Recreation OR Health Issues/Policies, and Nutrition OR Information OR Media OR Personal Characteristics and Beliefs OR Science OR Social Issues and Domestic Policy OR Technology OR Values
> **Exclude:** Experimental Questions
> **Exclude:** Open Ended Questions

Figure 6: Search string used to query the Roper iPoll database. **Bold** indicates field names, while all other text is field content. Keyword exclusions and topics were chosen to minimize personal experience questions because LLMs do not have personal experience histories about which to be uncertain.

- refused
- refused/web blank
- skip
- skipped
- skipped on web
- skipped/refused
- skipped/web blank
- web blank

After refusal filtering, remaining human answer choice ratios were re-normalized to sum to $100\%$. Questions with fewer than two answer choices after removal of refusal options, totaling 332 questions, were then removed. Finally, questions for which the total human response ratio sum was invalid were removed from consideration, totaling 572 questions. Valid answer ratio sum values are those that fall into the range $(100 - N, 100 + N)$, where $N$ is the number of answer choice options post-refusal-removal, representing the maximum divergence from 100 given worst-case rounding errors. The resulting set of 29667 questions were then sampled for 3000 questions uniformly at random without replacement. Two questions were found after experimentation time to have anomalous choice counts, likely due to those questions having more than 26 available options, which was not accounted for in the initial processing. These were removed and excluded from all analysis, leaving the reported 2998 questions. Due to licensing constraints, the final set of questions is not released in the supplementary materials.

## B ADDITIONAL FIGURES

This appendix contains the JSD shift permutation testing figures as described in Section 5.2 for all evaluated models. Dotted lines represent the JSD shift value for a given measure. All associated values are reported in Table 2.

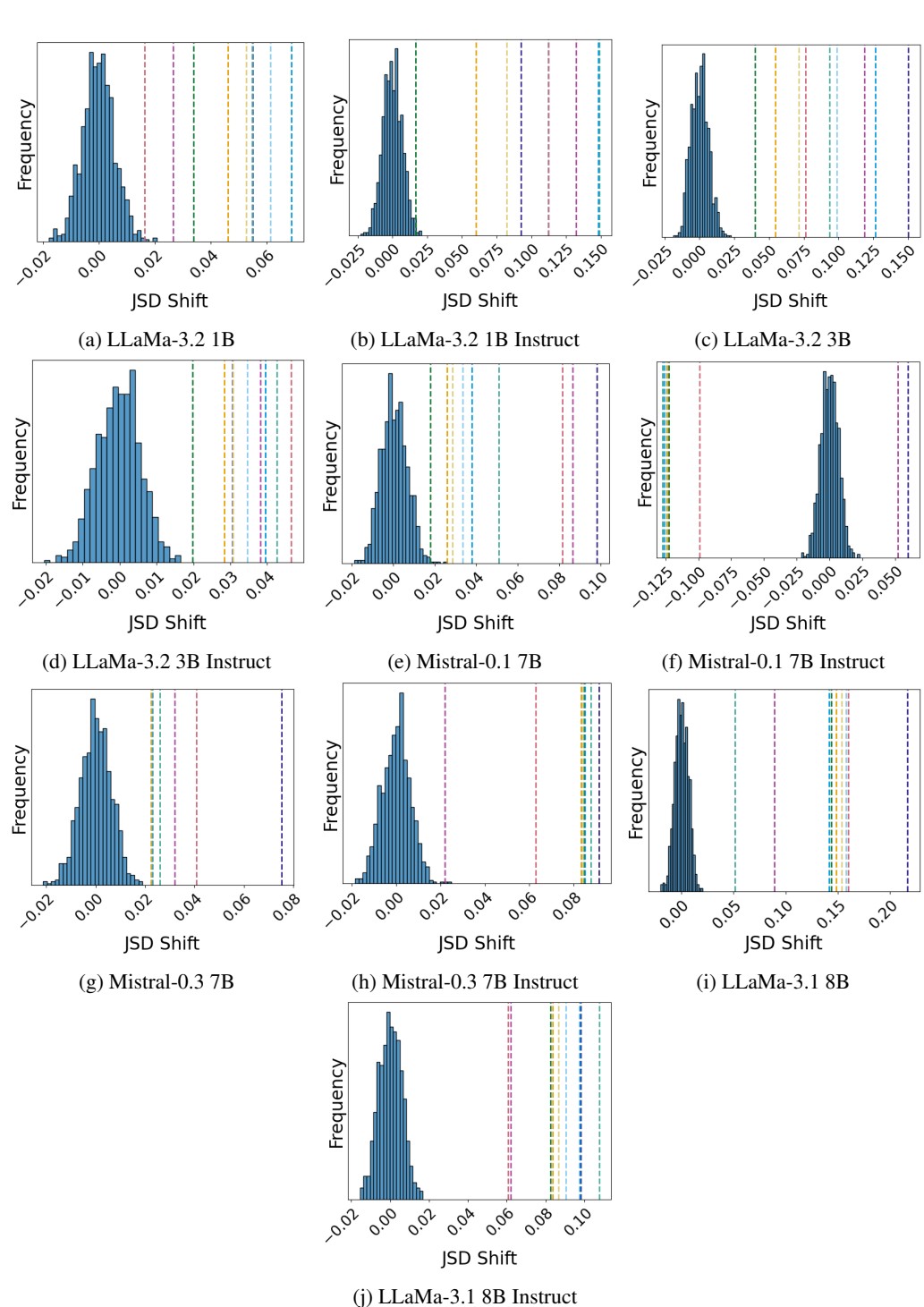

Figure 7: Jensen-Shannon Distance Shift permutation testing results for all models. Each dotted line is the observed JSD shift for a single uncertainty measure.

