# OpenReview forum: "Human-Alignment and Calibration of Inference-Time Uncertainty in Large Language Models"
_ICLR.cc/2026/Conference — Submitted to ICLR 2026_

### Official Review · Reviewer_6fia · 2025-10-28

**Soundness:** 2
**Presentation:** 2
**Contribution:** 2
**Rating:** 4
**Confidence:** 2

**Summary:**

How aligned are LLMs' uncertainty to people's judgements of uncertainty? In this work, the authors compare a people's uncertainty judgements to models' judgements. The authors propose and investigate several different measures of uncertainty calibration.

**Strengths:**

The authors take on an important problem --- how human-aligned are models' uncertainty. I really enjoyed and appreciate the authors' emphasis on compute-efficient measures of evaluation. I also appreciate the breadth of metrics the authors consider. The work is quite comprehensive. I enjoyed Figure 2 and would have loved to see more of a deep dive into its results (see below).

**Weaknesses:**

While I believe the paper has potential to be very strong, the current version was structurally challenging to follow. The motivation for the work (to my understanding) centered around evaluating models' uncertainty relative to people. However, this is only really done in Section 4. Section 5 is then not grounded in human data at all? I felt Section 5 came "out of the blue" and disrupted the story of the paper.

One idea would be to break up Fig 2 into more parts, expand Section 4, and substantially limit Section 5 in this piece. Or, flip the order so that the bulk of the emphasis is on the human evaluation. Structurally, the current paper is unfortunately highly confusing.

There are also no error bars in the main results, making it hard to assess how general and reliable the particular measures are.

**Questions:**

In addition to the questions/comments in my Weaknesses section:

- It’d be nice to show more on the human uncertainty, e.g., in the Appendix. Not exactly clear how MUCH uncertainty there is in the human data. Scatterplots, for instance, would be more revealing than bar graphs, e.g., a point for each query/trial (or at least a subset) to look at alignment.
- Were the survey datasets included in the LLM training data? They were all before 2023 so could have been? Does that influence uncertainty alignment? I realize it's out of scope for any rebutall period, but the paper would be much stronger with a new human eval as well to really assess alignment with models (for questions that are gaurenteed out of the models' training distribution).

---

> ### Author Response · Authors · 2025-12-02
> **Response to Reviewer 6fia**
>
> We thank the reviewer for their helpful review. We will attempt to address every provided weakness and question.
>
> We apologize for any lack of clarity in the purpose of the research. We will attempt to make sure the motivation is as clearly stated as possible in the final version. For clarity here, this work is equally focused on establishing both human-similarity in LLM uncertainty (called alignment herein) and calibration of model uncertainty to model correctness. The former has only minimal prior work (cited in the paper), while the latter is the focus of nearly all work on LLM uncertainty quantification. This work is intended to act as a bridge between these two foci, both strengthening the alignment results and establishing that some subset of the aligned uncertainty measures can also be useful in terms of the calibration usually sought after.
>
>
> We further appreciate the recommended changes and additions to the existing work. The human uncertainty statistics and representative visualizations, we agree, is a valuable addition that we will most certainly add to the final version. Because it is non-essential to the overall study purpose in spite of its value, we will add this as an additional appendix. Conversely, we do not believe that adding error bars to our existing figures is a realistic, or necessary, addition. Only figure 2 is conducive in form to adding error bars, but the already dense graphic will become only more inaccessible with the added complexity of error bars. Further, the data being represented therein (correlation between human uncertainty and LLM uncertainty, grouped by model and measure) does not provide a natural partitioning scheme to facilitate meaningful uncertainty measurement.
>
>
> Finally, the possibility of some subset of the human response data being present in the LLM training dataset(s) is, unfortunately, unknowable because the exact training data used by Meta and MistralAI are both proprietary information to which we do not have access. That being said, the fact that our data sources require login credentials to access reduces, but cannot eliminate, the probability of contamination. It should be noted that this concern is inherent to all studies on closed-training models, as even newly created contexts (with or without human comparison trials) may overlap significantly with the training data by pure happenstance, especially for simple tasks. We also considered and hope to follow-up this work with human trials in future work.

---

### Official Review · Reviewer_6yEC · 2025-10-30

**Soundness:** 2
**Presentation:** 2
**Contribution:** 1
**Rating:** 2
**Confidence:** 3

**Summary:**

The paper evaluates various LLM output distribution choices - entropy over the full vocab, top-k entropy, top-p entropy, and ‘choice’ entropy (entropy over the tokens corresponding to the answers in a finite set, e.g. a b c for MCQ questions) - for (i) human-alignment (agreement to human preferences and entropy), measured on a large survey dataset of US public opinion, and (ii) calibration, measured on MMLU.

The authors find that human alignment exists to a moderate degree at the top-token, but the ordering of non-top tokens does not correlate to human preference ordering. Furthermore, the degree of entropy of the models on each question correlates well in general with the entropy of human responses. Finally, the authors show that choice entropy has the best calibration on ECE, though all measures are moderately well calibrated.

Overall, the paper’s analysis suffers from several weaknesses detailed below; and the contribution, even if these were to be rectified, is marginal.

**Strengths:**

1. To my knowledge, this is the first work that extensively studies top-k and/or top-p of the logit-distribution in terms of calibration and human alignment.
2. A reasonable range of models is used in the experiments.

**Weaknesses:**

1. Regarding human alignment, the experimental design does not seem to be sound. The authors primarily use two datasets, which are public surveys conducted in the US, to gather human opinions. These datasets consist of questions asked over a wide range of dates (2017-2023), and are often on topical issues such as politics. Therefore, the human distribution is not stationary; however, all the LLMs tested are off-the-shelf open-source models, and therefore will have static cutoffs and remain stationary over the time period. Some questions will not make sense based on the cutoff in question, either due to the question already being resolved by the cutoff date, or because the premise of the question is not relevant at the time of the cutoff date yet.
2. Furthermore, the actual impact of the current design is limited. The current design only looks at the aggregate human distribution of responses and how closely LLM logit distributions align with these out of the box – and this is alignment turns out to be weak w.r.t. the top token. Instead, the authors could have directly stated a goal such as using LLMs to replace humans in population surveys (a line of work which does have prior art and interest); and then conducted interventions to try and improve the simple baseline scores reported (such as prompting variations, etc).
3. The paper does not provide examples of questions from the above human survey datasets, so I had to go track these down myself to find out what they look like.
4. Regarding calibration, the authors detail some motivation for why they don’t simply use out-of-the-box normalised entropy, but this should be conducted as an ablation rather than simply stated.
5. ‘Global’ entropy as used in the paper comes with the concomitant risk of distributional shift due to domain change, but this is not commented on nor experimented with by the authors.
6. The analysis of calibration is restricted to simply the MCQ setting (which the authors acknowledge), even though it is easily extensible to the open-ended setting. Furthermore, only a single dataset is used, MMLU (though I acknowledge that this single dataset consists of multiple subject topics), and so out-of-distribution analysis – an important element of calibration analysis – cannot be, and is not, done here.
7. The calibration section has no comparisons to baselines at all despite the plethora of previous works examining e.g. ECE on MMLU. It is therefore not clear what the intended contribution of this analysis is.
8. One of the contributions listed in the introduction is the claim that top-p sampling in LLMs is equivalent to the notion of the highest density credible set in Bayesian statistics. The authors state that this is “an important but previously un-noted connection between the fields”. However, there is no justification or exposition in the paper to support this statement. If interpreted at face value, the statement is also not much of a contribution - it is a fairly trivial insight.

**Questions:**

See weaknesses above.

---

> ### Author Response · Authors · 2025-12-02
> **Response to Reviewer 6yEC (1 of 2)**
>
> We thank the reviewer for their detailed review. We will attempt to address each of the identified weaknesses in the order presented.
> 1. The non-stationarity of the dataset is a concern we had during experiment design that, in the end, could not be effectively acted upon for the Roper dataset due to the sheer size of the dataset and the abundance of topical questions without clear identification criteria. We clearly state our cleaning methods in the appendix that are intended to remove questions and answer choices in part to address this issue to the extent possible procedurally. It is also stated in the primary inspiring work (we make it clear our Pew dataset comes from the authors of that work, not from Pew itself) that such topical questions were explicitly removed, addressing this concern there but with the secondary effect of severely reduced experiment power due to a small dataset. Further, on the assumption of the null hypothesis (no alignment), the instability that would be introduced by non-stationarity in the dataset sources would likely only dilute correlation rather than enforce it. As such, a false positive result, particularly at the correlation strengths and dataset size presented, is highly unlikely.
> 2. While we appreciate and recognize the value of the recommended extensions, we do not believe they are necessary additions to the current work beyond their mention as possible motivations. The work presented herein is intended only as pure research into the measurement of inference time uncertainty behavior of LLMs as compared to known human behavior that we hope will serve as a basis for the types of application research that were recommended.
> 3. As we stated clearly in the limitations (section 7) of the paper, we are prevented by existing copyright protections from releasing either the Pew or Roper datasets publicly. We worked to mitigate this by clearly detailing our data collection process in appendix A and providing clear links to the original repository. Our only options for providing example questions, given this legal limitation, are to clearly provide the prompting format (which we did in section 4.2) and to create a new example question, which runs the risk of misleading readers about the contents of the dataset.
> 4. We have run the requested additional analysis and will include the resulting figure to the appendix. Comparing with the normalized results, the majority of model-measure pairs remained relatively unchanged. The most noteworthy exceptions are the degradation in LLaMa 3.2 1B for some values of k (higher values indicate higher expected error). Despite this, every single measure average decreases when non-normalized, suggesting that the values reported originally were typically pessimistic compared with the naive approach. We have included a public link to the resulting data table here: https://imgur.com/a/ErjlGDo
>
> 5. We do not at any point in the work use the term “global entropy”, though we will infer for this answer that this was meant to refer to “total entropy”. Distributional shift of the token probabilities is expected across all domain and context variations, as is nearly definitional of any sufficiently complex language model. As such, we do not recognize this as a valid issue without further information.
> 6. We recognize the limited generalizability of studies performed exclusively within the MCQA setting, but we see this as initial research that will inspire and guide further research in a wider variety of more realistic and useful settings. We respectfully disagree without clear description how human-LLM UQ comparison is easily extensible to open-ended settings and we believe this will be a viable, but difficult endeavor in future work. On the point of the limits of using only MMLU, no pre-existing dataset can be guaranteed to be out-of-distribution in the meaningful sense (i.e. not present in the training data) and creation of a new dataset for calibration analysis is out of scope and an unreasonable ask for every single work. We believe that MMLU's ubiquity in LLM analysis and comparison should be sufficient for it to be considered a valid, if weak, dataset here.

---

> ### Author Response · Authors · 2025-12-02
> **Response to Reviewer 6yEC (2 of 2)**
>
> 7. As discussed in lines 287-291, normalized entropy may not necessarily correspond to the confidence percentage typically expected when calculating ECE, meaning that our ECE values may not be directly comparable to those reported in non-entropy-based UQ measures. As such, our ECE analysis focused primarily on providing additional comparison between measures, with raw calibration being more directly measured by the correlation and JSDS analyses. That being said, we investigated the literature further and found multiple studies which suggest our measured ECE values are competitive with similarly simplified tasks and experiment design. In particular, compare with the first row of table 3 in [1] and the ZSL (zero-shot learning) column of figure 4b in [2]. Both of these will be added as references to the ECE results discussion.
> We appreciate this point being noticed and brought to our attention. We agree that the connections between top-p and Bayesian credibility sets is not expounded upon sufficiently in the existing text, being a result of an oversight in consistency editing upon cutting for space. This connection was an auxiliary claim with only weak relation to the other claims of our paper and did not make the final cut. Given sufficient space, we hope to expound upon this claim in the final version of this work, but we agree with the point made here and will be removing it from our presented list of contributions.
>
> [1] Ye, Fanghua, et al. "Benchmarking llms via uncertainty quantification." Advances in Neural Information Processing Systems 37 (2024): 15356-15385.
>
> [2] He, Guande, et al. "Investigating uncertainty calibration of aligned language models under the multiple-choice setting." arXiv preprint arXiv:2310.11732 (2023).
>
> Note that [2], despite being a preprint, has been accepted as a poster presentation in an ICLR 2024 workshop (see https://openreview.net/forum?id=I1OcffsjUX).

---

### Official Review · Reviewer_rQzh · 2025-10-31

**Soundness:** 2
**Presentation:** 1
**Contribution:** 2
**Rating:** 2
**Confidence:** 4

**Summary:**

This paper investigates the alignment of inference-time uncertainty measures in large language models (LLMs) with both human group-level uncertainty and traditional calibration metrics. The authors evaluate a variety of entropy-based and probability-based uncertainty measures on a large dataset of survey questions. They find that several measures, particularly those based on entropy over various subsets of the token distribution (e.g., choice entropy, top-k entropy), show strong alignment with human uncertainty, even when the models' answer preferences do not align with human preferences. The paper also introduces a novel distributional calibration measure, the Jensen-Shannon Distance (JSD) shift, to assess how well model uncertainty predicts changes in the answer distribution. The results indicate that the human-aligned uncertainty measures are also moderately to strongly calibrated.

**Strengths:**

The core problem of evaluating whether an LLM's uncertainty corresponds to human uncertainty is important for building more transparent and trustworthy AI systems. The methodology is sound and described clearly. The creation and use of a large-scale dataset from the Roper Center is a significant contribution. The finding that uncertainty alignment can exist independently of preference alignment is a particularly interesting and non-obvious result.

**Weaknesses:**

The study's primary limitation is
1. The models used are mainly small open-sourced non-reasoning models. Experiment on more diverse and larger models, including closed-source SOTA models such as GPT-4, Gemini, Claude etc., would strengthen the claims, as uncertainty measures are more relevant in widely deployed SOTA models.
2. The method mainly focuses on multiple-choice questions. While this is a necessary simplification, the true test of these uncertainty measures will be in open-ended generation tasks. The proposed conceptual framework for extending the method to open-ended questions is a good first step, but it remains to be seen how effective it will be in practice.
3. The writing style can be improved for example, the contribution section in the introduction can be better formatted.

**Questions:**

Main:
1. The main question I have is, as I pointed out above, how well the findings generalize to larger and more diverse models beyond the small open-sourced models.

Minor
1. The paper finds a surprising lack of alignment in preference ordering between the models and humans, which contrasts with some prior work. The authors suggest this might be due to the cloze testing prompt format. Could the authors elaborate on this? Would CoT prompting or other prompt engineering techniques potentially improve preference alignment?
2.  The JSD shift results for the Mistral-0.1 7B Instruct model are intriguing, suggesting it might be "anti-calibrated." Do the authors have any hypotheses for why this specific model exhibits this behavior? Could it be an artifact of its instruction-tuning process?

---

> ### Author Response · Authors · 2025-12-02
> **Response to Reviewer rQzh**
>
> We thank the reviewer for their thorough review. We will attempt to address all the proposed weaknesses and questions in the order presented.
>
> A. Weaknesses
>   1. We recognize the possible limits to the generalizability in our results due to our choice of models, an issue that we must stress would be present regardless of the set of models studied. We hope that future work can expand into larger models and find ways to adapt the questions to black-box models, which is currently not feasible given our research question’s reliance on direct access to output logits. Regardless of future intended research, we believe the results reported here are significant enough for publication.
>   2. Limitation of our study to the MCQA domain is a simplification to facilitate an early foray into the question of UQ alignment, with the expectation that future work will expand into a wider variety of domains more similar to those experienced by the end user. This limitation, we should note, is clearly discussed at the beginning of section 7 (Limitations).
>   3. We appreciate any constructive criticism the reviewer can provide in terms of writing and formatting, but this weakness is difficult to address without specific examples. We will be happy to review and edit our language throughout.
>
> B. Questions
>
>   1. Please see the response to Weakness 1 above.
>   2. Counterfactual prompting, as described by the authors referenced in section 4.3, is an alternative prompting strategy that moves the “model’s” choice into the context and measures the model’s agreement with that choice based on its probability of responding with a representative token (e.g. “agree”, “correct”, etc. depending on format). The stated goal is that comparisons across various answer options will be less affected by differences in the prior probability of the text of the choices. Follow-up work by the same authors have found significant shifts in model behavior based on the chosen prompting strategy (counterfactual vs the more common cloze testing). We suspect that this might be because counterfactual prompting is less aligned with typical training methods, as it requires that the model must learn to more strongly prefer a specific token across multiple contexts, rather than one token above others in a single context. It is very possible that CoT or other prompt engineering methods may provide gains in preference alignment. This could be a great additional study as a follow-up to both our work and the work we cited. Preference order misalignment, however, is an auxiliary finding that serves only to contextualize the UQ alignment findings as indicative of alignment only in uncertainty level, not in the specifics of the preference distribution.
>   3. We agree that this is a great question, for which we unfortunately do not have a satisfying answer. Anti-calibration is an unfortunate and unexpected behavior, but it is difficult to isolate a specific probable cause. The afflicted model (Mistral 0.1 7B Instruct) does not uniquely possess any identifiable features that are not present in other models included in our study. All Mistral models use a relatively unique attention mechanism (sliding window attention) that could play a role, but this does not seem to have affected the other Mistral models. In addition, the model in question is neither the largest nor smallest model being considered. Our best hypothesis, based on the normal behavior of the non-instruct Mistral 0.1 7B model, is that the calibration behavior was unintentionally lost through some mechanism akin to abliteration during instruction fine-tuning. Unfortunately, this is difficult to verify, in part because this effect is not present in any other completion/instruct model pairing.

---

### Official Review · Reviewer_X4Jh · 2025-11-01

**Soundness:** 3
**Presentation:** 3
**Contribution:** 3
**Rating:** 6
**Confidence:** 4

**Summary:**

This paper investigates inference-time uncertainty measures for large language models and asks two questions: (i) how well such measures align with human group uncertainty (estimated via survey-response disagreement), and (ii) how well they are calibrated to correctness. Using roughly 3k multiple-choice items from Roper (plus a small subset from Pew) and several 1B–8B open-weight models (base and instructed), this paper compares families of token-probability signals, including top-1 probability, choice entropy, total entropy, top-k entropy, and top-p metrics. The study introduces a distributional calibration check based on Jensen–Shannon-distance shift, alongside per-subject Spearman correlations and ECE-style calibration. Main findings: multiple entropy-based measures—especially choice entropy and top-k variants—show moderate to strong alignment with human uncertainty across models; the same measures also show evidence of correctness calibration on MMLU; preference-order alignment remains weak even when top-answer agreement is above chance; a pathway is outlined for extending these signals to open-ended generation via a reduction to a three-way judgment.

**Strengths:**

- **Originality.** This paper explicitly targets **human-aligned** uncertainty (beyond correctness calibration) and connects **top-p** selection to Bayesian highest-density sets; it also proposes **JSD shift** as a distributional calibration diagnostic.
- **Quality.** Careful separation of **alignment** vs **calibration**; broad metric family; multi-model evaluation; subject-wise analyses; multiple complementary criteria (correlation, ECE, JSD shift).
- **Clarity.** Clear prompt template, dataset construction details, and heatmap/table visualizations that make the mixed “agreement vs ordering” story easy to parse.
- **Significance.** Practical, **low-overhead** signals (token-probability–based) that can drive runtime control/abstention in black-box deployments; highlights **choice entropy / top-k entropy** as strong default options.

**Weaknesses:**

- **Multiple-choice scope.** Evidence is limited to MCQ; the open-ended extension is conceptual and untested.
- **Prompt/decoding sensitivity.** Alignment differs from prior “counterfactual prompting” studies; results may depend on the **cloze** template and decoding choices.
- **Model coverage.** Only ≤8B open-weight models are included; larger/API models are referenced but not comprehensively stress-tested.
- **Metric/threshold choices.** Heuristic thresholds (e.g., |r|≥0.3) and standardization-based ECE binning may affect conclusions; alternatives could be reported.
- **Compute profiling.** End-to-end **latency/memory** impacts of computing metrics at scale are not fully quantified.
- **Failure analysis.** Notable negative/anti-calibration cases (e.g., a 7B instruct variant) merit deeper diagnosis.

**Questions:**

1. **Generalization to open-ended tasks:** Can you validate the proposed 3-way reduction empirically (few-shot rubric, judge variability), and compare against entailment-based or reference-guided scoring?
2. **Template & decoding effects:** How robust are alignment/calibration results to prompt variants, temperature, and nucleus/top-k settings?
3. **Combined signals:** Do simple fusions (e.g., choice entropy + top-k entropy) yield better Pareto fronts for risk–coverage/ECE?
4. **Cost accounting:** Please report wall-clock, memory, and throughput for per-token computations across models/subjects; include guidance for real-time use.
5. **Edge cases:** Analyze subjects/models with **anti-calibration** (positive correlation) to identify linguistic or dataset artifacts.
6. **Scale & APIs:** Replicate key plots on larger and API models; verify whether trends (choice/top-k entropy dominance) persist.
7. **Human side:** Report inter-survey reliability and how class-imbalance in human choices affects “alignment” correlations.

---

> ### Author Response · Authors · 2025-12-02
> **Response to Reviewer X4Jh - Weaknesses**
>
> We thank the reviewer for their thorough review. We will attempt to address all the proposed weaknesses and questions in the order presented.
>
> Weaknesses
>
> 1. Limitation of our study to the MCQA domain is a simplification to facilitate an early foray into the question of UQ alignment, with the expectation that future work will expand into a wider variety of domains more similar to those experienced by the end user, including open-ended domains. This limitation, we should note, is clearly discussed at the beginning of section 7 (Limitations).
> 2. LLMs are known to experience substantial prompt-sensitivity and we cannot guarantee generalization across a wide variety of prompts with the current experimental setup. That being said, we believe that the results reported herein are substantial, worth reporting, and defensible. In particular, we have two main points. First, the questions included in the Pew and Roper datasets are highly variable in their content and presentation. We did not remove redundant questions from the dataset, meaning that wording-varied questions from separate source surveys are present and will have naturally imposed some level of prompt variability. Second, and less importantly, we preselected our template before running any experimentation and did not, at any point, tune or vary the template. Given the large size of the roper dataset and the strength of the correlations show in figure 2, it is unlikely that the template chosen would perform anomalously and uniquely well compared with alternative valid prompts.
> 3. We recognize the possible limits to the generalizability in our results due to our choice of models, an issue that we must stress would be present regardless of the set of models studied. Our work was limited to models with <10B parameters by our available compute resources, though we hope and expect for future research to expand into larger models. We mitigated this concern to the extent possible by incorporating smaller models, which qualitatively showed little to no correlation between model size and alignment. We believe that these results warrant dissemination at the current suite of models.
> 4. Our choice of significance threshold in correlation analysis (|r| >= 0.3) is consistent with standard practice in behavioral studies and would have no effect on the numerics of our results, only their interpretation. For ECE analysis, we justified and explained the choice of binning methodology in section 5.1.1. That being said, we recognize the concern about potential obfuscation and will gladly calculate and provide ECE results with unnormalized confidence scores as an additional appendix.
> 5. We unfortunately do not have access to compute profiling for model inference at experiment time, as it was not tracked during experiment execution. All UQ methods that were investigated are cheap post-processing calculations that add little to no overhead, especially in comparison to the non-inference-time methods currently in vogue, which often require numerous rounds of generation to obtain a single uncertainty score. Our methods, importantly, are roughly constant in relation to model size, as the complexity of each UQ measure is dependent only on the model vocabulary size, which typically remains fixed across all sizes of a given model family.
> 6. We agree completely that the anti-calibration behavior seen in Mistral 0.1 7B Instruct across all tested UQ measures is both concerning and interesting, and it most certainly warrants further investigation. Given that this is isolated to a single model (including not applying to the non-chat-finetuned variant of the exact same model) and across all uncertainty measures, we would posit that this is indicative of a quirk of that particular model which deserves its own dedicated investigation. In either case, a deep diagnosis of failure on individual models is beyond the scope of this work.

---

> ### Author Response · Authors · 2025-12-02
> **Response to Reviewer X4Jh - Questions**
>
> Questions
>
> 1. Limitation of our study to the MCQA domain is a simplification to facilitate an early foray into the question of UQ alignment, with the expectation that future work will expand into a wider variety of domains more similar to those experienced by the end user, including open-ended tasks. This limitation, we should note, is clearly discussed at the beginning of section 7 (Limitations).
> 2. We have split our response here for clarity
>
>     a. LLMs are known to experience substantial prompt-sensitivity and we cannot guarantee generalization across a wide variety of prompts with the current experimental setup. That being said, we believe that the results reported herein are substantial, worth reporting, and defensible. In particular, we have two main points. First, the questions included in the Pew and Roper datasets are highly variable in their content and presentation. We did not remove redundant questions from the dataset, meaning that wording-varied questions from separate source surveys are present and will have naturally imposed some level of prompt variability. Second, and less importantly, we preselected our template before running any experimentation and did not, at any point, tune or vary the template. Given the large size of the roper dataset and the strength of the correlations show in figure 2, it is unlikely that the template chosen would perform anomalously and uniquely well compared with alternative valid prompts.
>
>     b. Many generation-time hyperparameters, most notably choices of p and k for top-k and top-p sampling, are not relevant to our experiments. Those hyperparameters limit the token output space and are redundant with many of the metrics described in the text. Those that are not redundant to top-p/k sampling would suffer from lack of experimental isolation.
>
>     c. The one significant counterexample to B.2.b above is the temperature setting. We intentionally did not vary the temperature (always using a default value of tau=1), given that its sole purpose is to modulate the entropy of the token probability distribution without altering relative ordering. This risks interference with our primarily entropy-based uncertainty metrics and obfuscation of the model’s uncertainty signal.
> 3. Our work did not investigate this question, as our research question was not intended to find an optimally calibrated UQ measure. The primary inspiring work (Moore et. al. 2025) incorporated signal fusion into their analysis of alignment to some success. We would hypothesize that fusion may yield better simultaneous alignment and calibration, but we have left this quesiton to future study.
> 4. I believe this was addressed in weakness 5 above.
> 5. Again, I believe this was addressed in weakness 6 above.
> 6. Addressed in weakness 3 above.
> 7. To our knowledge, inter-survey reliability cannot be viably assessed for our datasets using any methods with which we are familiar. The survey responses are obtained from an aggregated and anonymized repository that does not allow for reinterview-based consistency checks like test-retest or gross difference rate analysis. Further, consistency on similar questions even under the relaxation of allowing disjoint cohorts is not viable given the large question set and the variety of wordings that equivalent survey questions may take. That being said, the suggestions that analysis of the relationship between human side metadata (e.g. class imbalance, class count, etc.) and alignment level is a valuable addition that we will certainly investigate.

---

### Official Review · Reviewer_13Kp · 2025-11-02

**Soundness:** 2
**Presentation:** 2
**Contribution:** 1
**Rating:** 2
**Confidence:** 4

**Summary:**

**Summary**:
proposes to measure the alignment between various logit-based uncertainty quantification (UQ) methods and human group-level uncertainty, which is operationalized using entropy of population answer choices. Across 10 LLMs and 2000+ multiple choice questions of recent surveys, results reveal that most logit-based UQ methods are (linearly) strongly correlated with human uncertainty ($|\rho| geq 0.3$) – with choice entropy, total entropy and top-k approaches systematically ranking higher across models and datasets. In the MMLU dataset, analysis considering three different calibration metrics reveal that choice entropy is systematically better calibrated being more closely related to the correctness of the model.


**Contributions**:
- Novel perspective on the uncertainty quantification problem, focused on measuring model calibration with respect to human uncertainty (as opposed to correctness).
     - However, a similar problem is studied in Moore et al (2025), which first investigated the alignment of additional sampling measures (including sampling measures). The main difference seems to be the evaluation of both human alignment and calibration. Note that calibration and human alignment are measured separately (in two different settings)..
- Use of a Jensen Shannon Distance Shift as a measure of calibration.

**Strengths:**

- S1. Interesting angle on uncertainty quantification (UQ) research in LLMs, exploring whether the miscalibration of LLMs implies that models are in fact reflecting human uncertainty over answers.
- S2. Overall well-written and organized. Figure 3 provides a summary view over the different MMLU subjects.
- S3. Main results are backed by hypothesis testing (Figure 2 and Table 2).

**Weaknesses:**

- W1. **Limited novelty**: while the paper expands on the differences between prior work (in lines 82-88) it appears incremental (increasing number of datapoints and carrying calibration analysis). Perhaps the authors can highlight differences in findings or how their extended analysis to calibration differs from findings in prior work.
- W2. The definition of “inference-time” is too broad and not sufficiently motivated: the arguments provided to narrow the experiments to logit-based approaches are  also applicable to training-time approaches to UQ. As such, training-based approaches to model calibration could also be suitable approaches to be assessed in this work.
- W3. **Experiments are conducted exclusively in 3 multiple-choice formats**: Pew Research Center and Roper Center for Public Opinion Research (2025) surveys for measuring uncertainty alignment and MMLU to measure calibration. While there’s value in evaluating uncertainty alignment in multiple-choice settings, it does not necessarily generalize to more realistic settings where users interact with LLMs in open-ended generations.
- W4. Results are obtained using a single prompt: given LLMs’ sensitivity to slight changes in prompts, one may wonder about the generalization of these findings to different prompts (e.g., 0-shot vs few-shot prompting).
- W5. Measurement of alignment between UQ methods and human group-level uncertainty has limitations, since it does not consider differences in the ordering of the answer choices. (See Questions for more details).

**Questions:**

**Questions**:
1. Analysis of the human agreement is conducted using three metrics: one focusing on measuring choice selection agreement, another focusing on preference ordering alignment, and the third one focusing on the discrepancy in human uncertainty and model’s uncertainty.
      1.a.) Given the focus on measuring alignment between human group-level uncertainty and various UQ methods’ uncertainty, it is unclear why the first two metrics are necessary (the metrics are independent of any UQ method and instead fully rely on the model distribution). Could the authors motivate the relevance of including such analysis in the paper?
      1.b.) To measure uncertainty alignment between UQ methods and human uncertainty, the paper proposes to measure the linear correlation between the UQ method and the entropy of the human distribution per answer. While using entropy provides an aggregate view of human uncertainty it loses information about which answer choice the model is more confident about. For instance, for a multiple choice question with 4 answer choices, the two distributions have the same entropy but refer to two completely different settings: [0.0, 0.0, 0.25, 0.75],  [0.75, 0.25, 0.0, 0]. In other words, if I understand the setup correctly, the use of entropy as the proxy for human group-level uncertainty followed by pearson correlation may be an overoptimistic measure of alignment. This is further validated by lines 215-238 which state that models are only moderately aligned in the top-token but have different multiple choice ordering. Can the authors please clarify whether this is a problem and how they tackled it (e.g., post-hoc manual analysis)?
3. In Section 5.1. The paper mentions the use of Spearman correlation between binary correctness and UQ measures (lines 274-275). However, due to the discreteness of the correctness variable, my intuition is that this is an ill-suited metric. Can you motivate this metric choice?


**Clarity**:
1. The paper positions itself as “inference-time uncertainty in LLMs”, defining this class of methods as “measures that are able to be calculated at any time during generation, without additional auxiliary generations” and “inference time measures are uniquely useful [...] without significant added computation”. However, in my understanding such definition does not exclude training time approaches – approaches which are not discussed in this paper. I suggest the authors further clarify the definition or that consider training-time approaches, i.e., approaches that approach calibration through fine-tuning models.
2. Table 1 caption (page 4): add information about what bold faces mean.
3. Lines 272-277 mention the analysis being split in two phases but only one phase is discussed: the analysis using Spearman correlation.
4. Line 432 mentions “model itself may be unusually negatively calibrated”. Please explain what negatively calibrated implies, since in the original classification definition calibration values can only be between 0 and 1.


**Supporting arguments**:
1.  Lines 98 refer to the “limited or no inference-time capabilities [...] like self-reporting [...] and multi-inference consistency”. Please clarify what “inference-time” capabilities are and provide adequate citations to back such arguments.


**Typo**:
- Line 153: “surveys .” → “surveys.”
- Legend title in Figure 2: “Measures” → “Models”
- Missing legend in FIgure 5. It’s difficult to attribute each colored line to a different baseline.

**Suggestion**:
- Figure 2 is great! But super dense! One possible suggestion if you do consider this, is to select the best UQ method out of the top-k entropy ablations, the best for top-p entropy and best for top-p size and plot those. Leaving the other variations to the appendix.
- Formatting of Table 2 is off. I recommend formatting it like Table 1.

**Additional limitations**:
- The paper should also mention the fact that this analysis is only applicable to open-source models or closed-source APIs providing access to top-k and/or top-p access.

---

> ### Author Response · Authors · 2025-12-02
> **Response to Reviewer 13Kp - Weaknesses**
>
> We thank the reviewer for the detailed and helpful review. We will attempt to address all of the proposed weaknesses and questions in the order they were raised.
>
> Weaknesses
>
> 1. We regret that the reviewer believes our work to have limited novelty. We describe our novel contributions in lines 58-69 and (as the reviewer noted) lines 85-88. In particular, we expand upon and substantially strengthen the results of the only, to our knowledge, pre-existing work on human-LLM uncertainty alignment. The prior work relied on a very limited dataset that severely limited the statistical power of the experiments reported, necessitating more robust replication. While this is arguably incremental, we also include non-incremental contributions. Firstly, we are the first, to our knowledge, to directly link analysis of calibration and alignment. Secondly, as an auxiliary contribution, we also developed and deployed a novel calibration assessment methodology in the form of JSD shift that allows for distributionally-aware calibration analysis rather than relying on simple correlation with an indicator variable or arbitrary binning as in ECE.
> 2. It is not clear in what way the definition of “inference-time” as described in our work is too broad. On review, we believe that the usage is clear but could be more explicitly provided (Inference-time UQ measures are “measures of uncertainty that can be assessed using information available from the model at a given generation step without the need for additional model inferences (e.g. model logits, previously generated text, attention activation, etc.)”, as is more briefly described in lines 35-37. While we strongly agree that inference-time measures would likely have significant utility for model training, this is beyond the scope of the current work and is left to future work.
> 3. Limitation of our study to the MCQA domain is a simplification to facilitate an early foray into the question of UQ alignment, with the expectation that future work will expand into a wider variety of domains more similar to those experienced by the end user, including open-ended domains. This limitation, we should note, is clearly discussed at the beginning of section 7 (Limitations).
> 4. We agree that this is a potential limitation that should be added to our discussion of limitations in section 7. LLMs are known to experience substantial prompt-sensitivity and we cannot guarantee generalization across a wide variety of prompts with the current experimental setup. That being said, we believe that the results reported herein are substantial, worth reporting, and defensible. In particular, we have two main points. First, the questions included in the Pew and Roper datasets are highly variable in their content and presentation. We did not remove redundant questions from the dataset, meaning that wording-varied questions from separate source surveys are present and will have naturally imposed some level of prompt variability. Second, and less importantly, we preselected our template before running any experimentation and did not, at any point, tune or vary the template. Given the large size of the roper dataset and the strength of the correlations shown in figure 2, it is unlikely that the template chosen would perform anomalously and uniquely well compared with alternative valid prompts.
> 5. We hope that this is satisfactorily answered in the related questions below. As described briefly in section 4.1, we randomly shuffled the answer choice orderings for each question to mitigate label and positional biases.

---

> ### Author Response · Authors · 2025-12-02
> **Response to Reviewer 13Kp - Questions (1 of 2)**
>
> Questions (note, the numbering here does not reset for each subcategory, e.g. Clarity, Typo, etc.)
>
> 1. We’ve split this response up to address the subpoints as you labeled them.
>
>     a. Choice selection agreement and choice preference agreement analyses were included as reference points to compare more traditional questions of alignment with the less studied uncertainty alignment. They provide valuable information, most starkly in that they show that the different alignment types can, but need not necessarily, co-occur. As such, we cannot infer alignment from agreement (the closest non-objective analog to correctness) and vice versa.
>
>     b. In light of 1.a above, it should be stressed that the goal of this work is to investigate the level of uncertainty alignment specifically across a variety of measure-model pairs. The three alignment analyses suggest strongly that the differing types of alignment need not co-occur. As such, it is not problematic that the entropy-based measure of uncertainty loses information about the model’s preferred choice. This would only be relevant information if assessing total multivariate alignment, in which case both pieces of information can be used in concert. Further, we posit that entropy is a fine measure of uncertainty alignment precisely because it does not take into account the specific choice preference. Our question is, informally, whether the model is uncertain in the same situations in which a human would be uncertain. If we take uncertainty to indicate relatively equal preference for more than one available choice, the two example distributions the reviewer provided are equally certain about their individually preferred choice in that specific context, even if those preferred choices differ.
> 2. We chose Spearman correlation because it is a standard correlation measure for correlating ordinal categorical data (binary counts) with continuous data (UQ measures). It is possible that Pearson correlation (or the closely related point-biserial correlation) would be more appropriate and accurate measures, but they would assume the UQ measurements follow a normal distribution. This has not been theoretically supported, so we instead opted for the safer choice in Spearman’s.
> 3. I believe this was addressed in weakness 2 above.
> 4. Fixed. For clarity here, the bold values indicate the maximum values for each row.
> 5. We appreciate this error being brought to our attention. This is likely a case of sloppy editing when adding an additional analysis phase late in the research process. The section in question should, instead, be identifying and introducing three phases: ECE analysis, correctness correlation, and JSD shift. We will update the section accordingly.
> 6. The metrics may have an inverse relationship with the model compared to other metrics or the expected sign of the correlation. E.g. as the model is more likely to get the answer right, the uncertainty measure tends to increase rather than the expected decrease. All metrics investigated in our work should yield a negative correlation with correctness (i.e. increased uncertainty should yield lower accuracy), but the “anti-calibrated” model-measure pairs instead show positive correlation. This is contrary to our definition of a “well-calibrated uncertainty measure” in lines 265-267. To ensure clarity, we will ensure consistency in our nomenclature throughout, particularly preferring “anti-calibration” over “negative calibration”, which we used interchangeably.
> 7. We are unsure of the specific objection being raised here. Our usage of inference-time is defined on lines 35-37, though the specific wording “inference-time capabilities” rather than “inference-time computability” or similar may have been sloppy on our part. Alternatively, we could understand ambiguity in our wording leading to the incorrect understanding that “self-reporting” and “multi-inference consistency” are inference-time UQ methods. We will address both of these issues out of an abundance of caution.
> 8. Fixed.
> 9. Fixed.
> 10. The omission of the legend in figure 5 was a deliberate choice to maximize the information available. Adding a sufficiently legible legend would necessitate removing at least one subfigure from the main text. Each subfigure, however, serves a particular purpose. Specifically, they display, in order, the worst (non-anomalous) observed calibration, the best observed calibration, and the anomalous behavior of Mistral 0.1 7B Instruct. It is also important to note that Figure 5 is, in terms of information conveyed, redundant with table 2 and all line values can be inferred based on relative values of JSDS rows for the associated model. We recognize this can be communicated better in the figure caption and will do so.

---

> ### Author Response · Authors · 2025-12-02
> **Response to Reviewer 13Kp - Questions (2 of 2)**
>
> 11. Your positive feedback is greatly appreciated! We agree that the figure is somewhat busy and your suggestion would help declutter to a meaningful degree. However, we believe that the value of the additional information found in comparison across variations in k and p settings outweighs the possible legibility gains. Removal to an appendix may be inappropriate in this case because the visually discernable alignment (as well as the cross-model consistency) is necessary to explain the choice of measures included in the calibration experiments.
> 12. This weakness is inactionable without further information about what is off about the format of Table 2. The differing formats between the two tables is a deliberate choice due to the sheer density of Table 2. Table 1 uses the more visually appealing booktabs style, while Table 2 uses a modified variation of the standard Latex table style. We experimented with the booktabs style for table 2 during writing and found it to be severely detrimental to the table’s readability. If this point is deemed important, we propose that reformatting table 1 to be more similar to table 2 to be the preferred track.
> 13. We implied this specific limitation in the limitations section (section 7), but we agree that the implications of the limitation (specifically the need for access to model logits) can and will be made more explicit.

---

### Author Response · Authors · 2025-12-03
**Note to the Metareviewer**

Given the recent change in review process, we've elected to compile and provide a summary of the points raised across all the reviews below, with a short response to each. We have also responded in much greater detail to every point in our direct responses to the original reviews. The summary is provided across the following top-level comments and are split into three categories:

   I. points we believe are largely or fully addressed in the original text

   II. points that are not fully addressed, but will be addressed in the final version if accepted

   III. points that are not addressed (except as future work) in text, but are outside the scope of the current work.

All points include references, of the form (ReviewerID - W#, Q#,...), that indicate which reviewer-identified weakness(es) (W) and question(s) (Q) are being addressed.

---

> ### Author Response · Authors · 2025-12-03
> **Summary - Addressed in the text (1 of 2)**
>
> I. Addressed in the text
>
> A. Concerns about novelty (13Kp - W1)
>
>     Our work is, to our knowledge, the second work to investigate uncertainty alignment in LLMs in any capacity. The only known previous work is not known to have undergone peer review and relies on a dataset that is too small for strong statistical power. This work substantially improves the robustness of the results by compiling and experimenting with a much larger dataset, establishes a bridge between uncertainty alignment and the much more commonly researched uncertainty calibration, and investigates a more targeted but more finely varied set of measures by experimenting exclusively on a larger swath of entropy-based measures.
>
> B. Definition and justification of Inference-time focus (13Kp - W2,Q3,Q7)
>
>     While we defined our usage of inference-time methods (lines 35-37), we will add additional wording to ensure clarity. We explicitly limit our analysis to this domain because inference-time uncertainty has the highest natural utility for human interpretability and as a training/reasoning signal.
>
> C. Limitation to MCQA domain (13Kp - W3; X4Jh - W1, Q1; rQzh - W2; 6yEC - W6)
>
>     Limitation of our study to the MCQA domain is a simplification to facilitate an early foray into the question of UQ alignment, with the expectation that future work will expand into a wider variety of domains more similar to those experienced by the end user, including open-ended domains. This limitation, we should note, is clearly discussed at the beginning of section 7 (Limitations)
>
> D. Model choice and generalizability concerns (13Kp - Q13; X4Jh - W3,Q6; rQzh - W1,Q1)
>
>     We disclose this limitation briefly in section 7, though we will add additional details. Our experiments are indeed limited to open source models (due to the necessity of logit access) below 10B parameters (due to hardware resource constraints). We hope to address both where possibly in future work, but the strength of our alignment results warrant dissemination.
>
> E. Dataset creation and cleaning methods (6yEC - W1)
>
>     We provide a detailed account of the Roper dataset creation process in Appendix A. Details of the Pew dataset creation are provided in the original work from which it was obtained. While the latter explicitly omitted survey questions that would be expected to show high inter-survey variability, ours does not. While this was largely due to viability concerns because of the increased dataset size, we posit their inclusion would be expected to degrade the correlation and are thus strengthen our positive results.
>
> F. Randomization and ordering effects (13Kp - W5)
>
>     As detailed in Appendix A, we did randomize answer choice orderings to avoid well-established ordering effects
>
> G. Relevance of choice and preference ordering alignment analysis (13Kp - Q1)
>
>     Choice and preference ordering alignment analysis were included because they are the more commonly studied notions of human-similarity. They therefore provide an important comparison point for establishing uncertainty alignment, particularly in that uncertainty alignment can diverge strongly from the other alignment measures across the same set of contexts.

---

> ### Author Response · Authors · 2025-12-03
> **Summary - Addressed in the text (2 of 2)**
>
> H. Release of Pew/Roper datasets (6yEC - W3)
>
>     As made explicitly clear in both section 8 (Reproducibility Statement) and Appendix A, we are legally forbidden from providing the datasets publicly due to copyright licensing restrictions. We have detailed our dataset creation process in Appendix A and provided the necessary code in our supplementary materials to fully recreate the Roper dataset. Pew dataset details are left to the original source paper, cited in text.
>
> I. Possibility of training data overlap (6fia - Q2)
>
>     As our data was obtained from online resources that predate the training of tested models, none of which publicly disclose their data source list, we could never guarantee the source studies do not appear in the training corpus. Given the much weaker alignment on top choice and preference ordering than uncertainty alignment, it is highly unlikely that our results could be primarily explained by data spoilage. This is one of many reasons why a follow-up involving novel human studies would be valuable, but beyond scope here.
>
> J. Definition and interpretation of “anti-calibration” and “negative calibration” (13Kp - Q6)
>
>     We identified a model (Mistral 0.1 7B instruct) that shows miscalibration in direct contrast to the desired behavior in terms of the sign of uncertainty/correctness correlation and Jensen-Shannon distance shift across most UQ measures. We termed this degenerate behavior as “anti-calibration” (though we inconsistently use the interchangeable “negative calibration”, which we will correct in the final version).
>
> K. Computational requirements and profiling (X4Jh - W5,Q4)
>
>     We did not perform computation profiling during our experiments due to the overall lack of relevance to the primary questions. To the extent that they are relevant (particularly in their relation to exclusively inference-time analysis), our methods are far cheaper than the state-of-the-art calibrated UQ measures, which often depend on multiple parallel model generation calls.
>
> L. Correlation error bars (6fia - W3)
>
>     Our analysis does not lend itself to the inclusion of error bars, as there are no natural binning schemes available aside from model selection. We sidestep this, however, by only advancing models that showed significant (in terms of correlation strength) alignment across ALL models
>
> M. (Inter/Intra)survey reliability (X4Jh - Q7)
>
>     It is not clear what benefit would be gained by establishing high (or even low) intersurvey reliability across the various surveys from which our Roper dataset is ultimately sourced. Classical methods that rely on individual follow-ups are impossible here, while simple per-subject cross-survey disagreement, as addressed in I.E above, would only serve to degrade the observed correlations.

---

> ### Author Response · Authors · 2025-12-03
> **Summary - Not currently addressed in text, but will be addressed for the camera-ready version**
>
> II. Not currently addressed in text, but will be addressed for the camera-ready version
>
> A. Inappropriate claimed contribution - top-p + Bayesian confidence interval equivalence (6yEC - W8)
>
>     This is a major oversight on our part that was resulted from removal of the relevant text for space-saving purposes without editing the contribution list appropriately. The offending claimed contribution was a minor auxiliary finding that has only tangential relevance to the remainder of the work and could be safely removed. The final version will absolutely have this contribution removed from the contribution list, which we posit is otherwise sufficiently compelling.
>
> B. Prompt Template diversity concerns (13Kp - W4; X4Jh - W2,Q2a)
>
>     LLMs are known to have strong behavioral shifts from small changes in context and this is a reasonable limitation that should and will be added to our disclosed limitations. An ideal version of the experiments herein would include varied prompt templates, but the strong alignment signal given no prompt tuning and substantial question content variance suggests, we posit, that the results would hold, even if slightly weakened, given extended analysis.
>
> C. Choice of statistical tests (13Kp - Q2; X4Jh - W4)
>
>     Most of our analysis relies on standard correlation analysis, with the exception of ECE (which is a standard measure in UQ research) and Jensen-Shannon Distance shift (which we introduce and justify in text). We chose Spearman correlation over Pearson correlation because we have no guarantee of normalily of uncertainty distribution. We will justify this explicitly in text in the final version.
>
> D. Non-normalized ECE results (6yEC - W4)
>
>     This was a valuable criticism that was considered and argued in text (section 5.1.1 specifically). For full transparency, we have re-run our ECE analysis on the non-normalized uncertainty scores. Our results show a slight improvement when non-normalized and will be included as an additional Appendix in the final version. The relevant table is linked here: https://imgur.com/a/ErjlGDo
>
> E. ECE baseline comparison (6yEC - W7)
>
>     This is another valuable ask that we overlooked in our initial analysis and reporting. We have identified and referenced ECE analysis on comparable tasks already available in the peer-reviewed literature. We will include these in the text of section 5.2.
>
> F. Hyperparameter choices (X4Jh - Q2b,Q2c)
>
>     Most hyperparameter choices of concern (particularly top-p and top-k sampling during generation) are actively avoided because they are redundant with and actively impede most of our uncertainty measures, which require varying portions of the entire token probability distribution. The primary exception, temperature, is not varied because its sole  purpose is to control the entropy of the output probability distribution, which also impedes all entropy-based measures.
>
> G. Roper dataset metadata and statistics (X4Jh - Q7; 6fia - Q1)
>
>     While not required to support the results of our work, we agree with the value in adding simple figures showing the distributions of uncertainty, answer choice count, etc. for the Roper dataset human responses. We will be adding this information as an additional appendix.
>
> H. Various minor figure/typo errors (13Kp - Q4,Q5,Q8,Q9,Q10,Q11,Q12; rQzh - W3; 6fia - W1,W2)
>
>     All of these with sufficient information to be actionable have been fixed.

---

> ### Author Response · Authors · 2025-12-03
> **Summary - Not addressed in text and outside of paper scope**
>
> III. Not addressed in text and outside of paper scope
>
> A. Potential for alignment/calibration improvement via fine-tuning (13Kp - W2)
>
>     We recognize the value of inference-time uncertainty as a signal during training and inference, with the former being a primary inspiration. We also recognize that, even with our strong alignment signals, we would likely be able to induce stronger alignment with uncertainty-aware finetuning. Both of these interpretations of the requested addition, however, are far outside of the scope of detecting the presence of uncertainty alignment in out-of-the-box models across various reasonable candidate measures.
>
> B. Investigation of anomalous behavior in Mistral 0.1 7B Instruct (X4Jh - W6,Q5; rQzh - Q3)
>
>     The anomalous anti-calibration behavior of this specific Mistral version is an interesting auxiliary finding for which we have no strong explanation given that it has no identifiable unique feature (or even combination of features) relative to our set of tested models. We agree with the reviewers that this warrants further investigation, but we posit that this deeper and targeted analysis is well outside this paper’s scope.
>
> C. Alternative prompting strategies, e.g. counterfactual, CoT, etc. (rQzh - Q2)
>
>     Similar to I.C above, alternative tasks and prompting strategies across those tasks may yield substantially different uncertainty and uncertainty alignment behaviors. We strongly support future work that explores the numerous avenues, with this work serving as the primary basis of comparison.
>
> D. Metric fusion (X4Jh - Q3)
>
>     We would hypothesize that combinations of our various uncertainty measures (as well as others not included herein) may yield more optimal simultaneous alignment and calibration. Prior work cited in the text has partially explored this question to limited success. We posit that a full treatment of the potential of metric fusion for uncertainty alignment also warrants a dedicated follow-up.
>
> E. Efficacy on downstream applications (6yEC - W2)
>
>     We agree strongly that uncertainty alignment holds potential benefits to numerous downstream applications, especially those involving some degree of human-AI cooperation/collaboration. This work establishes the necessary precepts for such follow-up application research.

---

### Meta-Review · Area_Chair_2fJK · 2025-12-25

**Summary:**

This paper studies the alignment of model uncertainty and human uncertainty. Through reviewers agree that it can be an interesting topic to study, concerns remain on the soundness of the empirical study, the limitation of multiple-choice tasks and small-scale models, and the novelty.

**Reviewer Concerns:**

1. limited novelty, appears incremental compared to related works;
2. experiments are conducted exclusively on multiple-choice tasks;
3. only small open-source non-reasoning model is used;
4. writing;
5. soundness of the experiment: human distribution is not stationary;

Overall, the author's rebuttal mainly contains arguments and promises. I think most concerns still remain.

**Reviewer Scores:**

Original rating is (2,6,2,4,2). I think reviewers would retain their rating.

---

### Decision · Program_Chairs · 2026-01-26

Reject